# Uptake of appointment spacing model of care and associated factors among stable adult HIV clients on antiretroviral treatment Northwest Ethiopia

Abaynew Assemu Asrade[1], Nurilign Abebe Moges[2], Maru Meseret[3], Kasaye Demeke Alemu[1], Tilahun Degu Tsega[4], Pammla Petrucka[5,6], Animut Takele Telayneh[2]*

1 HIV/AIDS Care Program, International Center for AIDS Care Program, Bahir Dar, Ethiopia, 2 Department of Public Health, College of Health Sciences, Debre Markos University, Debre Marqos, Ethiopia, 3 Department of Health Informatics, College of Health Sciences, Debre Markos University, Debre Marqos, Ethiopia, 4 College of Health Sciences, Injibara University, Injibara, Ethiopia, 5 College of Nursing, University of Saskatchewan, Saskatoon, Canada, 6 School of Life Sciences and Bioengineering, Nelson Mandela African Institute of Science and Technology, Arusha, Tanzania

* animuttakele@gmail.com

**Data Availability Statement:** The datasets used and/or analyzed during the current study are available as supporting information.

## Abstract

### Introduction

Ethiopia launched an Appointment Spacing Model in 2017, which involved a six-month clinical visit and medication refill cycle. This study aimed to assess the uptake of the Appointment Spacing Model of care and associated factors among stable adult HIV clients on ART in Ethiopia.

### Methods

A cross-sectional study was conducted from October 3 to November 30, 2020 among 415 stable adult ART clients. EpiData version 4.2 was used for data entry and SPSS version 25 was used for cleaning and analysis. A multivariable logistic regression model was fitted to identify associated factors, with CI at 95% with AOR being reported to show the strength of association.

### Results

The uptake of the appointment spacing model was 50.1%. Residence [AOR: 2.33 (95% CI: 1.27, 4.26)], monthly income [AOR: 2.65 (95% CI: 1.13, 6.24)], social support [AOR: 2.21 (95% CI: 1.03, 4.71)], duration on ART [AOR: 2.41 (95% CI: 1.48, 3.92)], baseline regimen change [AOR: 2.20 (95% CI: 1.02, 4.78)], viral load [AOR: 2.80 (95% CI: 1.06, 7.35)], and alcohol abstinence [AOR: 2.02 (95% CI: 1.21, 3.37)] were statistically significant.

### Conclusions

The uptake of the ASM was low. Behavioral change communication, engaging income-generating activities, and facility-level service providers' training may improve the uptake.

**Funding:** The author(s) received no specific funding for this work.

**Competing interests:** The authors have declared that no competing interests exist.

## Introduction

The Human Immunodeficiency Virus (HIV) has continued as a major public health problem. Globally, 38 million people living with HIV, among 2.2 million new infections, and 690,000 deaths in 2019 [1]. The African continent contributes to about 11% of death in the global population-related HIV [2]. Ethiopia is among the African countries significantly affected by the HIV pandemic [3]. By the end of 2019, in Ethiopia, 525,921 HIV-positive adults know their HIV status and of the HIV-aware individuals, 97.1% were receiving Anti-Retroviral Treatment (ART) [4]. Although Ethiopia is close to reaching HIV epidemic control [4], treatment failure remains a major challenge due to discontinuation of ART, and poor ART adherence. Furthermore, patients endure long traveling distances, high transportation costs, long waiting times to get ART service, persistent HIV-related stigma and discrimination, poor attitudes by healthcare workers, and high caseloads within ART facilities which contribute to overall poor ART service in Ethiopia [2, 5, 6].

According to WHO recommendation 2016, the innovative service delivery of care has been developed to accommodate the increasing number of stable individuals on ART and improve retention in care and health outcomes. Ethiopia endorsed and started implementing this new service delivery model of care Appointment Spacing Model (ASM) in April 2017 [5, 7, 8]. ASM was designed for clinically stable adults receiving ART patients to reduce the number of clinic visits, offload the workload of a healthcare facility to improve adherence and quality of care address patient needs, reduce travel costs, and decrease waiting times to care, which resulted improve self-management and retention [5, 8–13] This approach was intended to reach 70% of stable adult clients living with HIV by offering the opportunity to have twice-yearly clinical visits for medication refill [5]. The target was set by considering the socio-cultural situation, degree of awareness, stigma, discrimination, resource demands, and the sustainability of the program [8]. This program has also a dual purpose for the patients to enhance patients health outcomes to achieve the three 90s targets set by the Joint United Nations Program on HIV/AIDS (UNAIDS), which aspires to have 90% of HIV-positive individuals know their status, with 90% receiving sustained ART, and 90% on ART achieving viral load suppressions (VLS) and health facility allow to expand access to HIV services [5, 6, 8, 9, 11, 12].

ASM is associated with the probability of death in which those patients enrolled in the program were less likely to die compared to their counterparts [14, 15] Although access to ART has significantly increased in recent years due to the current test and treat strategy, poor ART adherence and retention have become a current and persistent public health concern [7]. Hence, ASM has been introduced to accommodate the increased demand for access to HIV services, improve the quality of care, and enhance treatment outcomes for HIV patients [5, 7, 8]. Since its introduction in Ethiopia, the number of eligible patients who enrolled in the ASM program has remained unacceptably below the target [16]. Factors associated with the uptake of ASM are socio-demographic factors, ART-related knowledge, and risk behaviors factors, clinical, and health service delivery-related factors that have been identified [5, 7, 10, 17–22]. In Ethiopia, there is a paucity of evidence on the uptake of ASM after introducing the service. Therefore, this study assessed the uptake of ASM among stable adult HIV clients on ART and identify its associated factors in select health facilities in Ethiopia.

## Materials and methods

### Study design and eligibility criteria

A facility-based cross-sectional study was conducted from October 3 to November 30, 2020. This study was conducted in 12 selected health facilities in the East Gojjam zone, Amhara

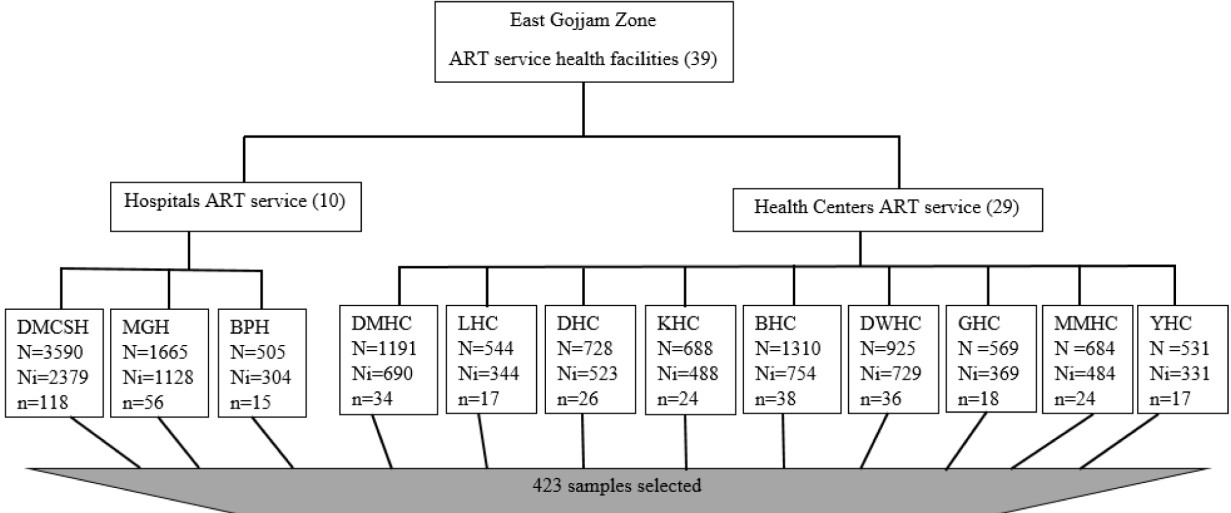

Note: ART- Anti Retroviral Therapy, DMCSH: Debre Markos Comprehensive Specialized Hospital, MGH: Mota General Hospital, BPH: Bichena Primary Hospital, DMHC: Debre Markos Health Center, LHC: Lumamie Health Center, DHC: Dejen Health Center, KHC: Kuye Health Center, BHC: Bichena Health Center, DHC: Debre Work Health Center, GHC: Gindewoin Health Center, MMHC: Mertolemariam Health Center, and YHC: Yejubie Health Center.
N = All ART enrolled patients, Ni = Adult stable on ART patients, and n = Samples proportionally allocated from adult stable on ART patients.

**Fig 1. Schematic presentation of sampling procedures for uptake of ASM among stable clients on antiretroviral therapy in East Gojjam Zone, Amhara, Northwest Ethiopia, 2020.** Note: ART-Anti Retroviral Therapy, DMCSH: Debre Markos Comprehensive Specialized Hospital, BPH: Bichena Primary Hospital, DMHC: Debre Markos Health Center, LHC: Lumamie Health Center, DHC: Dejen Health Center, KHC: Kuye Health Center, BHC: Bichena Health Center, DHC: Debre Health Center, GHC: Gindewoin Health Center, MMHC: Mertolemariam Health Center, and YHC: Yejubie Health Center. N = All ART enrolled patients, Ni = Adult stable on ART patient, and n = Samples proportionally allocated from adult stable on ART patients.

region [23]. Currently, thirty-nine public health facilities; ten hospitals, and twenty-nine health centers provide ART services in the zone. During the study period, more than eighteen thousand people were living with HIV and all ART rendering facilities were providing ASM services in the study area [23]. Among these ART sites, three hospitals, and nine Health Centers were high caseload ART facilities that hosted more than 500 cases selected for this study in the Zone [23, 24]. Stable adult clients on ART were considered as a source population and patients accessing care in high caseload ART health facilities were potentially included in the study (**Fig 1**).

## Sample size determination and sampling procedures

The sample size was determined using a single population proportion formula by considering 50% proportion as there is no previous evidence on uptake of ASM, 5% level of precision, 95% confidence level (CI), and 10% non-response rate yielding a final sample of 423. Based on this sample calculation, all study participants were selected from high caseload ART facilities through proportional allocation to each facility based on their respective patient loads during the study period. Finally, each study participant was selected using a systematic random sampling technique of the "K" interval for every 3rd patient.

## Definition of variables

**Uptake of ASM.** Eligible ART clients who fulfill the World Health Organization's (WHO) criteria of stable adult clients accepting the six months schedule for both clinical evaluation and medication refill were categorized as **'uptake'** and coded as "1" whereas those who refused to be enrolled in ASM were categorized as **'non-uptake'** and coded as "0" [6].

**Stable adult clients.**   Patients who are on ART and meeting the following criteria were considered stable: at least one year, aged $\geq$18 years old, no adverse drug reactions requiring regular monitoring, a good understanding of lifelong adherence, have two consecutive viral loads <1000 copies/ml or CD4 counts above 200 cells/mm$^3$, no acute opportunistic infections, and not pregnant or breastfeeding [5].

**Alcohol use.**   Patients were scored on the Cut down, Annoyed, Guilty, and Eye-opener (CAGE) Substance Abuse Screening Tool which was scored 0 for "no" and 1 for "yes" answers, with a higher score being an indication of alcohol problems. A total score of $\geq$2 is considered clinically significant [17].

**Adherence to HIV chronic care principles.**   A tool on chronic care assessment of the **5As** (assess, advise, agree, assist, and arrange) was used. Responses were scored ranging from 5–20. A summed score greater than the mean value is considered "acceptable care", otherwise the individual was assessed as "not acceptable care" [18].

**Social support.**   The Oslo social support scale (OSSS-3) consists of three items (Oslo 1: score range 1–4; Oslo 2: scores range 1–5; Oslo 3: scores range 1–5). The sum score can be operationalized into three broad categories of social support: poor 3–8, moderate 9–11, and strong 12–14 [19].

**ART-related knowledge.**   Each respondent's knowledge about ART (6 items) was scored and summed. One point was given for each question that was answered correctly. Participants' level of knowledge was treated as good if they scored correctly on more than half of the ART-related knowledge questions [20].

## Data collection tools, procedures, and quality assurance

The data collection tool was prepared in English and then translated into Amharic. All data were translated to English to maintain consistency and coherence for analysis. Data were collected using exit interviews supported with individual chart review techniques. The tools were prepared after reviewing prior research as well as the Ethiopian ART intake and follow-up forms [5, 7, 10, 17–22]. Twelve nurse data collectors and four public health professional supervises participated. To ensure data quality, training was given to both data collectors and supervisors. Close supervision was maintained during the entire data collection period. All filled questionnaires were checked for completeness, clarity, and consistency. Any missed or unfilled data was corrected immediately during the data collection period. Finally, all collected data were reviewed and checked for completeness before data entry.

## Data processing and analysis

Collected data were coded and entered using EpiData Version 4.2 and exported to SPSS Version 25 software for data cleaning and analyses. Both bi-variable and multivariable logistic regression models were fitted. Variables with p-values <0.25 in the bi-variable analysis were selected for multivariable analysis. Model fitness was checked through the use of the Hosmer-Lemeshow test. Descriptive statistics computed included mean, median, and standard deviation were presented using frequency tables, figures, and charts. CI at 95% with Adjusted Odds Ratio (AOR) was used to identify statistically associated factors for the uptake of ASM.

## Ethics statement and consent to participate

All the procedures in the present study were approved by the ethics committee in research at Debre Markos University, College of Health Science (HSC/R/C/Ser/Co/56/11/13). Written informed consent was obtained from each study participant before initiating the study.

Confidentiality of the information was maintained. All methods were performed in accordance with the relevant guidelines and regulations.

## Results

### Socio-demographic, economic, and service delivery-related factors

A total of 415 clients on ART participated in this study with a response rate of 98%. The median age of study participants was 39 ± 13 years. Females constituted 248(59.8%), most of the study participants 254(52.5%) were married, and 167(40.3%) of them had no formal education (**Table 1**).

### ART knowledge and risk behaviors related factors

One hundred (24%) of the study participants had a good level of knowledge related to ART. The majority (92.5%) of study participants reported ART consists of drugs that suppress the activity of HIV. A significant number of the respondents presented a lack of understanding of technical

**Table 1. Socio-demographic, economic, and service delivery-related factors of uptake ASM among stable clients on antiretroviral therapy in East Gojjam Zone, Amhara, Northwest Ethiopia, 2020 (n = 415).**

| Variables | Characteristics | Frequency N (%) |
|---|---|---|
| Age in years | 18–24 | 22(5.3) |
| | 25–34 | 116(28.0) |
| | 35–44 | 154(37.1) |
| | ≥45 | 123(29.6) |
| Monthly income | ≤2500 birr | 278(67.0) |
| | 2501–5000 birr | 77(18.6) |
| | ≥5001 birr | 60(14.4) |
| Religion | Orthodox | 356(85.8) |
| | Muslim | 48(11.6) |
| | Protestant | 11(2.6) |
| Marital | Single | 63(15.2) |
| | Married | 218(52.5) |
| | Divorced | 90(21.7) |
| | Widowed | 44(10.6) |
| Educational status | No formal education | 167(40.3) |
| | Primary (1–8 grades) | 127(30.6) |
| | Secondary (9–12 grades) | 76(18.3) |
| | College and above | 45(10.8) |
| Partner educational status (256) | No formal education | 105(41.0) |
| | Primary (1–8 grades) | 81(31.6) |
| | Secondary (9–12 grades) | 41(16.0) |
| | College and above | 29(11.4) |
| Occupation | Employed | 335(80.7) |
| | Unemployed | 80(19.3) |
| Health facility type | Health Center | 230(55.4%) |
| | Hospital | 185(44.6%) |
| ART facility catchment | Within the catchment | 294(70.8) |
| | Without the catchment | 121(39.2) |
| ART clinic access & maintain its own privacy | No | 90(21.7) |
| | Yes | 325(78.3) |
| Travel distance to ART facility | ≤60 minute | 323(77.8) |
| | >60 minute | 92(22.2) |

**Table 2. Knowledge and risk behaviors related factors of uptake ASM among stable clients on antiretroviral therapy in East Gojjam Zone, Amhara, Northwest Ethiopia, 2020 (n = 415).**

| Variables (Expected answer) | Characteristics | Frequency N (%) |
|---|---|---|
| ART consists of drugs that cure HIV/ADIS (no) | No | 304(73.3) |
| | Yes | 111(26.7) |
| ART consists of drugs to suppress the activity of HIV(yes) | No | 31(7.5) |
| | Yes | 384(92.5) |
| CD4 count is the number of HIV viruses in the blood (no) | No | 311(74.9) |
| | Yes | 104(25.1) |
| Viral load is the number of HIV viruses in the blood (yes) | No | 45(10.8) |
| | Yes | 370(89.2) |
| ART increases the viral load (no) | No | 345(83.1) |
| | Yes | 70(6.9) |
| ART increases the CD4 count (yes) | No | 291(70.1) |
| | Yes | 124(29.9) |
| Alcohol use problem | No | 286(68.9) |
| | Yes | 129(31.1) |
| Frequency of condom use during sexual intercourse | Always | 77(18.6) |
| | Sometimes | 117(28.2) |
| | Never | 221(53.3) |
| Number of the sexual partner in the last 6 months | Had no sexual intercourse | 174(41.9) |
| | One sexual partner | 200(48.2) |
| | ≥2 sexual partner | 41(9.9) |
| Who is your sexual partner (241) | Husband/wife | 194(80.5) |
| | Commercial sex worker | 17(7.1) |
| | Bare ladies | 9(3.7) |
| | Unknown him/her self | 21(8.7) |
| Ever had chew khat | No | 372(89.6) |
| | Yes | 43(10.4) |

terms regarding CD4 counts. Above half (53.3%) of respondents never used condoms and about one-third of participants were identified as having problems with alcohol use (**Table 2**).

## Clinical care-related factor

In this study, the uptake of ASM was 208(50.1%) with a gender difference (19% of males and 31.1% of females). Almost half (49.6%) of the study participants had documented hemoglobin (Hgb) levels during their follow-up appointments. The majority (88%) of participants experienced a change from their baseline regimen. In this study, one in three HIV patients ever missed his/her clinical visit, the main reasons given were forgetting (37.1%), too busy (36.4%), sickness (10.7%), shortage of transport cost (7.9%), family member sickness (5%) and other conditions (2.9%). Nearly 94% of participants elected to the uptake of ASM to reduce the frequency of facility visits. Besides, 80% of study participants not uptake ASM cited personal preference as the reason (**Table 3**, **Figs 2 and 3**).

## Factors affecting uptake of appointment spacing model of care among stable adult ART patients

In this study, the bi-variable analysis, variables like facility type, age, monthly income, marital status, educational status, occupation, discloser status, social support, traveling distance, ART

**Table 3. Clinical care-related factors of uptake ASM among stable clients on antiretroviral therapy in East Gojjam Zone, Amhara, Northwest Ethiopia, 2020 (n = 415).**

| Variables | Characteristics | Frequency N (%) |
|---|---|---|
| HIV chronic care status | Not acceptable care | 167(40.2) |
|  | Acceptable care | 248(59.8) |
| Duration on ART | 13–60 months | 155(37.3) |
|  | ≥61 months | 260(62.7) |
| Current WHO clinical stage | I | 391(94.2) |
|  | II | 24(5.8) |
| Recent viral load | <250 RNA copies/ ml | 386(93.0) |
|  | ≥250 RNA copies/ ml | 29(7.0) |
| Recent CD4 count (282) | <500 cell/mm$^3$ | 35(47.9) |
|  | ≥500 cell/mm$^3$ | 147(52.1) |
| Baseline regimen | TDF+3TC+NVP | 60(14.5) |
|  | AZT+3TC+NVP | 93(22.4) |
|  | AZT+3TC+EVF | 22(5.3) |
|  | TDF+3TC+DTG | 13(3.1) |
|  | TDF+3TC+EFV | 227(54.7) |
| Current regimen(365) | TDF+3TC+DTG | 345(94.5) |
|  | Other* | 20(5.5) |
| INH prophylaxis status | Not complete | 52(12.5) |
|  | Completed | 363(87.5) |
| CPT status | Ongoing taking | 68(16.4) |
|  | Complete | 239(57.6) |
|  | Not given at all | 108(26.0) |
| Ever missed a clinical appointment | No | 275(66.3) |
|  | Yes | 140(33.7) |
| Willingness to uptake appointment spacing model of ART care | No | 196(47.2) |
|  | Yes | 219(52.8) |

Note

* ABC+3TC+EFV, AZT+3TC+DTG, and TDF+ 3TC+EFV.

facility catchment, ART clinic cleanness and maintain privacy, condom use, ever khat chew, sexual partner, duration of ART use, viral load, and baseline regimen change with p-value <0.25 were selected for multivariable analysis. In multivariable analysis patients living out of their ART catchment area were 2.33 times more likely to uptake ASM compared to those living within the catchment area [AOR: 2.33 (95% CI: 1.27, 4.26)]. Study participants' monthly income ≥5001 birr were more than 2.5 fold times as likely to uptake ASM compared to counterparts [AOR: 2.65 (95% CI: 1.13, 6.24)]. ART patients with strong social support were 2.21 times more likely to uptake ASM compared to ART patients who have poor social support [AOR: 2.21 (95% CI: 1.03, 4.71)]. Similarly, HIV patients with ≥ 61months duration on ART were 2.41 times more likely to uptake ASM compared to those with less time using ART [AOR: 2.41 (95% CI: 1.48, 3.92)]. HIV patients with baseline regimen change had 2.2 times the uptake ASM compared to those who not change their baseline regimen [AOR: 2.20 (95% CI: 1.02, 4.78)]. Study participants with viral load <250 RNA copies/ml showed 2.8 fold more apt to the uptake of ASM compared to counterparts with high viral load levels [AOR: 2.80 (95% CI: 1.06, 7.35)]. Lastly, HIV patients who did not use alcohol were 2 times more likely to uptake the ASM compared to alcohol-using HIV patients [AOR: 2.02 (95% CI: 1.21, 3.37)] (Table 4).

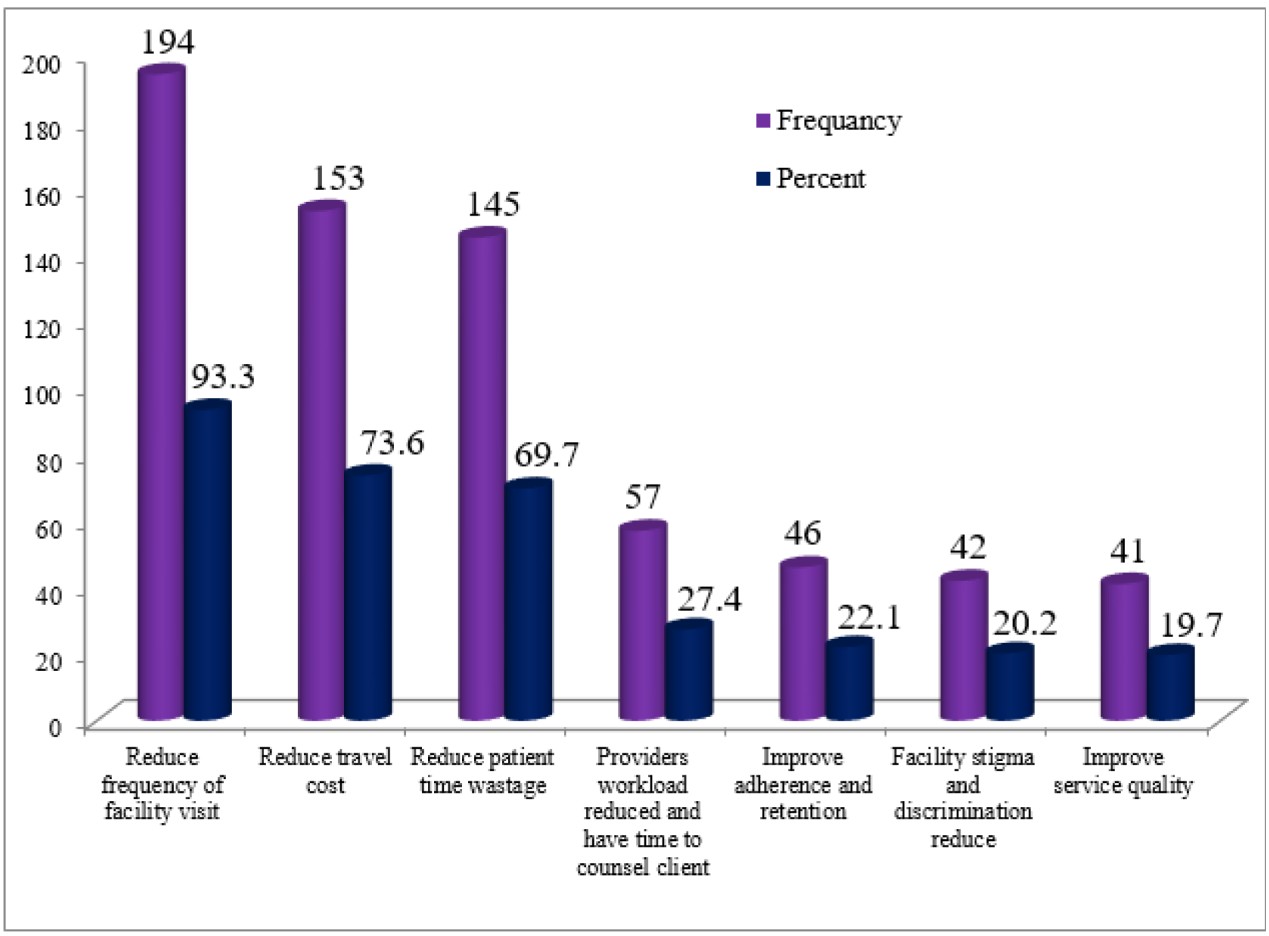

**Fig 2. Reasons for uptake of ASM among stable clients on antiretroviral therapy in East Gojjam Zone, Amhara, Northwest Ethiopia, 2020 (n = 208).**

## Discussion

This study investigated the proportion of ASM uptake and associated factors among stable adult clients on ART in Ethiopia. In this study, the proportion of uptake of ASM for HIV care was 50.1% (95% CI 45, 55) with higher rates in female than male study participants. This uptake is higher than similar studies reported from 7.2% in Uganda [25], 10.3% in Zambia [26], and 28% in South Africa [27]; however, it lower than previous evidence from Guinea 59.6% [10] and 69% [12] in Malawi. This variance may reflect the difference in countries' ASM care eligibility criteria for enrolment in a model. For example, in Malawi, people living with HIV including children greater than 2 years old, adolescents, adults, and specific populations, who are well, in care for three or more months, and have suppressed viral load, are eligible for ASM of care [28]. Whereas in some countries, including Ethiopia, with low coverage of viral load testing, supply chain concerns, or other systems challenges, simpler national guidelines are offered that prescribe a combined clinical and refill visit every six months for every adult patient only [28]. Another reason for the discrepancy might be the difference in facility type, key clinical values, CD4 count, and viral load at the entry to care, as well as attributes of study settings.

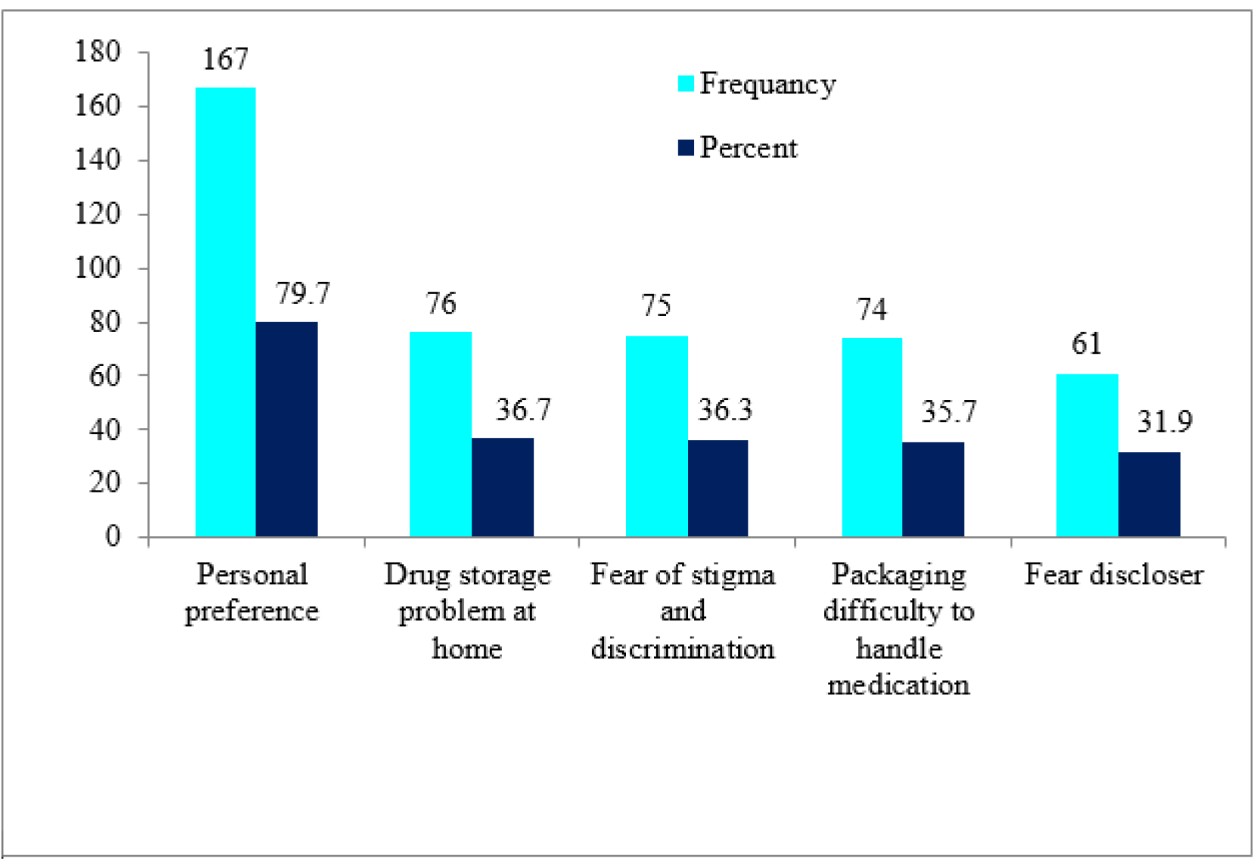

**Fig 3. Reasons for not uptake ASM among stable clients on antiretroviral therapy in East Gojjam Zone, Amhara, Northwest Ethiopia, 2020 (n = 207).**

In this study, those patients who reside outside the catchment area of their ART facilities were 2.33 times more likely to uptake ASM than those who resided within the catchment area. The major reason given was that those clients outside their locality (out of the catchment area) incur both direct and indirect costs. Hence, ASM was found to save their working time and travel costs. This finding is consistent with studies conducted in Uganda, Rwanda, and Zambia [25, 29–31]. This may be due to travel distances to get ART service, inconvenient transportation, long facility wait times for patients and their families, and COVID-19 pandemic lockdown were possible factors impacting the uptake of ASM. Despite the distance from health care being positively associated with ASM uptake, it is recommended to avail HIV treatment services in the nearby facilities and encourage clients to use the local health facilities. The finding implied that HIV patients should be advised to attend regular follow-ups in nearby health facilities.

This study also revealed that patients whose household monthly income ≥5001 birr were 2.65 times more likely to uptake ASM. The probable reasons for a good level of ASM acceptance by high-income groups may be related to these patients having higher-level incomes being employed, engaged in income-generating activities, having convenient drug storage, reducing working time lost due to frequent health facility visits, and experiencing better access information from different sources. This result does not coincide with the study conducted in Kenya, Cameroon, and Namibia [32–34], which may reflect the country-level differences in available infrastructure.

**Table 4. Bi-variable and multi-variable analysis uptake of ASM among stable clients on antiretroviral treatment, East Gojjam Zone, Northwest Ethiopia, 2020 (n = 415).**

| Variable | ASM status | | COR with | AOR with | P- |
|---|---|---|---|---|---|
| | Uptake | Not uptake | 95% CI | 95% CI | Value |
| **Health facility type** | | | | | |
| Health center | 104 | 126 | 1 | 1 | |
| Hospital | 104 | 81 | 1.55(1.05, 2.09) | 1.52(0.93, 2.50) | 0.09 |
| **Age in years** | | | | | |
| 15–24 | 11 | 11 | 0.64(0.25, 1.59) | 1.34(0.43, 4.21) | 0.61 |
| 25–34 | 48 | 68 | 0.45(0.27, 0.76) | 0.68(0.35, 1.31) | 0.25 |
| 35–44 | 74 | 80 | 0.59(0.36, 0.96) | 0.72(0.40, 1.27) | 0.25 |
| ≥45 | 75 | 48 | 1 | 1 | |
| **Monthly income** | | | | | |
| ≤2500 birr | 134 | 144 | 1 | 1 | |
| 2501–5000 birr | 35 | 42 | 0.89(0.54, 1.48) | 1.09(0.59, 2.01) | 0.78 |
| ≥5001 birr | 39 | 21 | 1.99(1.11, 3.56) | **2.65(1.13, 6.24)** | **0.025** |
| **Marital status** | | | | | |
| Single | 26 | 37 | 1 | 1 | |
| Married | 112 | 106 | 1.50(0.85, 2.65) | 1.93(0.82, 4.50) | 0.13 |
| Divorced | 44 | 46 | 1.36(0.71, 2.60) | 1.54(0.66, 3.57) | 0.31 |
| Widowed | 26 | 18 | 2.05(0.94, 4.48) | 1.28(0.45, 3.59) | 0.64 |
| **Educational status** | | | | | |
| No formal education | 90 | 77 | 1 | 1 | |
| Primary (1–8 grades) | 54 | 73 | 0.63(0.39, 1.00) | 0.59(0.34, 1.03) | 0.06 |
| Secondary (8–12 grades) | 39 | 37 | 0.90(0.52, 1.55) | 0.96(0.49, 1.88) | 0.91 |
| College and above | 25 | 20 | 1.07(0.55, 2.07) | 0.57(0.21, 1.50) | 0.25 |
| **Occupation** | | | | | |
| Employed | 163 | 172 | 1 | 1 | |
| Not employed | 45 | 35 | 1.35(0.83, 2.22) | 1.50(0.82, 2.75) | 0.19 |
| **HIV Disclosure status** | | | | | |
| No | 16 | 28 | 1 | 1 | |
| Yes | 192 | 179 | 1.88(0.98, 3.58) | 1.48(0.68, 3.23) | 0.33 |
| **Social support** | | | | | |
| Poor | 95 | 112 | 1 | 1 | |
| Moderate | 79 | 72 | 1.30(0.86, 1.98) | 1.14(0.69, 1.89) | 0.60 |
| Strong | 34 | 22 | 1.82(1.00, 3.35) | **2.21(1.03 4.71)** | **0.041** |
| **Travel distance to ART site** | | | | | |
| ≤60 minutes | 151 | 172 | 1 | 1 | |
| >60 minutes | 57 | 35 | 1.85(1.15, 2.98) | 1.24(0.63, 2.43) | 0.53 |
| **ART facility catchment** | | | | | |
| Within the catchment | 129 | 165 | 1 | 1 | |
| Without the catchment | 79 | 42 | 2.40(1.55, 3.73) | **2.33(1.27, 4.26)** | **0.006** |
| **ART clinic access and ability to maintain the privacy** | | | | | |
| No | 53 | 37 | 1.57(0.99, 2.52) | 1.67(0.95, 2.94) | 0.08 |
| Yes | 155 | 170 | 1 | 1 | |
| **Alcohol use problem** | | | | | |
| No | 159 | 127 | 2.04(1.34, 3.13) | **2.02(1.21, 3.37)** | **0.007** |
| Yes | 49 | 80 | 1 | 1 | |
| **Frequency of condom use** | | | | | |

(*Continued*)

**Table 4.** (Continued)

| Variable | ASM status | | COR with | AOR with | P- |
| --- | --- | --- | --- | --- | --- |
| | Uptake | Not uptake | 95% CI | 95% CI | Value |
| Always | 34 | 43 | 1.28(0.72, 2.29) | 0.68(0.34, 1.35) | 0.27 |
| Sometimes | 59 | 58 | 1.37(0.81, 3.21) | 1.58(0.88, 2.83) | 0.12 |
| Never | 115 | 106 | 1 | 1 | |
| **Number of the sexual partner in the last six months** | | | | | |
| No sexual partner | 94 | 80 | 1.83(0.92, 3.67) | 1.95(0.76, 5.03) | 0.164 |
| One(husband /wife) | 98 | 102 | 1.50(0.75, 2.98) | 1.04(0.40, 2.68) | 0.94 |
| ≥2 sexual partner | 16 | 25 | 1 | 1 | |
| **Ever had khat chew** | | | | | |
| No | 192 | 180 | 1.80(0.94, 3.45) | 1.22(0.54, 2.75) | 0.63 |
| Yes | 16 | 27 | 1 | 1 | |
| **Duration of ART** | | | | | |
| 13–60 months | 54 | 101 | 1 | 1 | |
| ≥61 months | 154 | 106 | 2.72(1.80, 4.12) | **2.41(1.48, 3.92)** | **0.001***|
| **Viral load** | | | | | |
| <250copies/ml | 199 | 187 | 2.36(1.05, 5.32) | **2.80(1.06, 7.35)** | **0.037** |
| ≥250copies/ml | 9 | 20 | 1 | 1 | |
| **Change baseline regimen** | | | | | |
| No | 14 | 36 | 1 | 1 | |
| Yes | 194 | 171 | 2.92(1.52, 5.59) | **2.20(1.02, 4.78)** | **0.045** |

Note

* p-value <0.001.

Findings from this study revealed that those patients who had strong social support were 2.21 times more likely to uptake the ASM compared to those who had poor social support. It is consistent with studies conducted in South Africa, the USA, and Sub-Saharan Africa [35–37]. Stronger social relationships are a rigorous protective factor against morbidity and all-cause mortality, and the supplemental guide on the ASM of HIV service delivery recommended these clients need additional support. The support can be enhanced through counseling to disclose to their family members and treatment supporters because the care and support provided by family members and communities were reported to boost self-worth that promotes positive coping.

Patients who did not have alcohol use were 2 times more likely to uptake the ASM than those who were alcohol used. This finding aligned with a study conducted in sub-Saharan Africa and East Africa [38, 39]. The finding indicated that alcohol use prevention mechanisms should be strengthened and incorporated into routine HIV care.

The Virological result had an association with ASM uptake. The current study indicated that HIV patients who had <250 RNA copies/ml were 2.8 times more likely to uptake the ASM compared to their counterparts. The finding is consistent with other studies conducted in Guinea, Zambia, and other Sub-Saharan African countries [10, 31, 40]. Patients who have a viral load result that cannot be detected (less than 250 copies/ml of viral load) indicated a good level of both drug and clinical adherence. Therefore, clients should be encouraged to continue to take their medicine as prescribed to keep the virus undetectable. The present study revealed the ASM enrollment criteria also triggered a low level of viral load. However, the effect of ASM on Virological suppression needs further study.

Months on ART were also associated with the level of ASM uptake. HIV patients with more than five years duration of ART were 2.41 times more likely to uptake ASM compared to their counterparts. This finding is consistent with previous studies conducted in Nigeria, the United States of America, and Ethiopia [16, 41, 42]. A possible explanation might be that patients receiving ART drugs for more than five years had adequate ART medication-related knowledge and skills, as well as were more likely to have disclosed to their family members and others in their community.

Another finding revealed that HIV patients who changed their first-line HIV drug regimen were 2.2 times more likely to uptake ASM compared to those who have not changed their baseline regimen. This finding concurred with studies done in Guinea, Malawi, and Nigeria [10, 41, 43]. The possible justification might be the increased pretreatment resistance to Neverapine resulted in poor treatment outcomes. Currently, the WHO recommends using other alternatives of Dolutegravir (DTG) base as a first-line regimen. Hence, most stable adult clients after changing their first-line regimen might have improved health outcomes related to improving adherence due to reducing the frequency of dosing. The majority of this study participants have changed their baseline regimen to Dolutegravir based this might be more likely to uptake ASM.

### Limitation of the study

Due to the nature of the cross-sectional study, we could not establish the causal relationship between the independent and dependent variables. Additionally, the uptake of ASM was measured without considering the duration of time after at least one year on ART. This may affect the proportion of ASM uptake over time. Hence, further study is recommended to measure ASM uptake over an extended time period.

### Conclusions

The proportion of the ASM for ART care uptake was low (50.1%) which was below the expected target that was 70%. Factors associated with the uptake of the ASM of antiretroviral treatment of care were socio-economical, behavioral, and clinical care-related. The association between ASM uptake and Virological suppression should be evaluated. Further, a cohort study is recommended to rule out the association between time and service uptake.

### Supporting information

**S1 Data. Appointment spacing model of care data sets among stable clients on antiretroviral therapy in East Gojjam Zone, Amhara, Northwest Ethiopia, 2020.**
(SAV)

### Acknowledgments

We would like to thank Debre Markos University, College of Health Sciences for their unreserved support, East Gojjam Zone Health Department, respective Health Facilities and their administrative staff, data collectors, and supervisors for their invaluable input to this work.

### Author Contributions

**Conceptualization:** Abaynew Assemu Asrade.

**Data curation:** Maru Meseret, Tilahun Degu Tsega.

**Formal analysis:** Abaynew Assemu Asrade, Maru Meseret, Kasaye Demeke Alemu, Tilahun Degu Tsega, Animut Takele Telayneh.

**Investigation:** Abaynew Assemu Asrade.

**Methodology:** Abaynew Assemu Asrade, Nurilign Abebe Moges, Kasaye Demeke Alemu, Pammla Petrucka, Animut Takele Telayneh.

**Software:** Abaynew Assemu Asrade, Maru Meseret, Tilahun Degu Tsega, Animut Takele Telayneh.

**Supervision:** Abaynew Assemu Asrade, Nurilign Abebe Moges, Maru Meseret, Kasaye Demeke Alemu, Tilahun Degu Tsega.

**Validation:** Nurilign Abebe Moges, Kasaye Demeke Alemu, Pammla Petrucka, Animut Takele Telayneh.

**Visualization:** Nurilign Abebe Moges, Pammla Petrucka.

**Writing – original draft:** Abaynew Assemu Asrade, Nurilign Abebe Moges, Maru Meseret, Animut Takele Telayneh.

**Writing – review & editing:** Abaynew Assemu Asrade, Nurilign Abebe Moges, Maru Meseret, Kasaye Demeke Alemu, Tilahun Degu Tsega, Pammla Petrucka, Animut Takele Telayneh.

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
