## [Decision Letter · Decision Letter 0]

2 Aug 2022

PONE-D-22-11061Uptake of Appointment Spacing Model of care and associated factors among stable adult HIV clients on antiretroviral treatment Northwest EthiopiaPLOS ONE

Dear Dr. Telayneh,

Thank you for submitting your manuscript to PLOS ONE. After careful consideration, we feel that it has merit but does not fully meet PLOS ONE’s publication criteria as it currently stands. Therefore, we invite you to submit a revised version of the manuscript that addresses the points raised during the review process. Can you please address the concerns raised by the expert reviewer?

We look forward to receiving your revised manuscript.

Kind regards,

Avanti Dey, PhD

Staff Editor

PLOS ONE

Journal Requirements:

Reviewers' comments:

Reviewer's Responses to Questions

**Comments to the Author**

1. Is the manuscript technically sound, and do the data support the conclusions?

Reviewer #1: Yes

2. Has the statistical analysis been performed appropriately and rigorously? 

Reviewer #1: Yes

3. Have the authors made all data underlying the findings in their manuscript fully available?

Reviewer #1: Yes

4. Is the manuscript presented in an intelligible fashion and written in standard English?

Reviewer #1: Yes

5. Review Comments to the Author

Reviewer #1: This is an important study about uptake of ART. The manuscript is well-written and has a nice layout, which make it easy to read and follow. However, I have a few concerns, which if considered and applied by the authors might improve its readability.

1. It’s unclear how appointment spacing model solves the challenges in line #53-57, it would be helpful for the authors to expound on this relationship in the text.

2. Related to the above comment #1, the authors state that ASM is associated with lower probability of death and use citation #7 to back that assertion, but that citation did not have anything to do with that finding. What is the relevance of this citation to this text? Given that observation, it now becomes unclear in the text, what is the benefit of ASM to ART, and why it should be advocated for people on ART. The authors need to cite an original source that developed the ASM model and the study that showed its benefits to people on ART. That will form the foundation and validity of this current study/investigation.

3. How were the three out of the 10 hospitals and nine out of 29 health centers selected for the study?

4. On line #91 the authors reported that “high caseload ART facilities which hosted more than 500 cases selected for this study”, but on line #97 they reported that “final sample of 423”.

5. The tables have n=415, the authors should consider revising the results section and the rest of the manuscript to reflect that sample size. The number of study participants should be consistent across the manuscript. Right now it is a little bit confusing as indicated above.

6. PLOS authors have the option to publish the peer review history of their article (what does this mean?). If published, this will include your full peer review and any attached files.

Reviewer #1: **Yes: **Chris B. Agala

---

## [Author Response · Author response to Decision Letter 0]

18 Sep 2022

Response to Reviewers

We are happy for your constructive comments and suggestions. We are also learned a lot from these substantial scientific comments to improve the quality of this manuscript. We are addressed your comments and suggestions as stated below. The manuscript was revised as indicated in the highlights and track change.

Journal Requirements:

Tank you! We are accepted your comment. We are prepared according to the journal guidleine and checked all refernces. We found only one “Cawley C, Nicholas S, Szumilin E, Perry S, Quiles IA, Masiku C, Wringe A: Six-monthly appointments as a strategy for stable antiretroviral therapy patients: evidence of its effectiveness from 7 years of experience in a Medecins Sans Frontieres supported programme in Chiradzulu district, Malawi. In: Journal of the International AIDS Society: 2016: John Wiley & Sons Ltd the Atrium, Southern Gate, Chichester, 2016” is identifed as retracted and removed from this list.

Comments to the Author

1. Is the manuscript technically sound, and do the data support the conclusions?

Reviewer #1: Yes

Thank you!

2. Has the statistical analysis been performed appropriately and rigorously?

Reviewer #1: Yes

Thank you!

3. Have the authors made all data underlying the findings in their manuscript fully available?

Reviewer #1: Yes

Thank you!

4. Is the manuscript presented in an intelligible fashion and written in Standard English?

Reviewer #1: Yes

Thank you!

5. Review Comments to the Author

Reviewer #1: This is an important study about uptake of ART. The manuscript is well-written and has a nice layout, which make it easy to read and follow. However, I have a few concerns, which if considered and applied by the authors might improve its readability.

1. It’s unclear how appointment spacing model solves the challenges in line #53-57, it would be helpful for the authors to expound on this relationship in the text.

Thank you! We are accepted your constructive comments and included your comment in the document page 3 lines 58-65.

2. Related to the above comment #1, the authors state that ASM is associated with lower probability of death and use citation #7 to back that assertion, but that citation did not have anything to do with that finding. What is the relevance of this citation to this text? Given that observation, it now becomes unclear in the text, what is the benefit of ASM to ART, and why it should be advocated for people on ART. The authors need to cite an original source that developed the ASM model and the study that showed its benefits to people on ART. That will form the foundation and validity of this current study/investigation.

Thank you! We are accepted your scientific comment and corrected as your comment in page 3-4 lines 58- 78.

3. How were the three out of the 10 hospitals and nine out of 29 health centers selected for the study?

Thank you! In this study, 30% and above health facilities providing ART service to ensure the representativeness and select high case load health facilities served more than 500 HIV patients in East Gojjam Zone were included. 

4. On line #91 the authors reported that “high caseload ART facilities which hosted more than 500 cases selected for this study”, but on line #97 they reported that “final sample of 423”. 

Thank you! All health facilities included in this study, >500 HIV patients enrolled and received ART. The sample size calculated for this study was 423 and this sampled study participants were proportionally allocated to each selected health facilities among stable adult ART patients. We are included the schematic presentation of sampling procedure (figure 1) in the document. 

5. The tables have n=415, the authors should consider revising the results section and the rest of the manuscript to reflect that sample size. The number of study participants should be consistent across the manuscript. Right now it is a little bit confusing as indicated above.

Thank you! In this study, 415 ART patients were participated in this study with 98% response rate. The remaining study participant were not welling to complete all the questionnaire components resulting 8 study participants were incomplete data and not included in the analysis.

6. PLOS authors have the option to publish the peer review history of their article (what does this mean?). If published, this will include your full peer review and any attached files.

Do you want your identity to be public for this peer review? For information about this choice, including consent withdrawal, please see our Privacy Policy.

Reviewer #1: Yes: Chris B. Agala 

Thank you for your constructive comments and suggestions‼!

---

## [Editor Report · Decision Letter 1]

14 Dec 2022

Uptake of Appointment Spacing Model of care and associated factors among stable adult HIV clients on antiretroviral treatment Northwest Ethiopia

PONE-D-22-11061R1

Dear Dr. Animut Takele Telayneh,

We’re pleased to inform you that your manuscript has been judged scientifically suitable for publication and will be formally accepted for publication once it meets all outstanding technical requirements.

Kind regards,

Hans-Uwe Dahms, Ph.D.

Academic Editor

PLOS ONE
---

## [Editor Report · Acceptance letter]

20 Dec 2022

PONE-D-22-11061R1 

Uptake of Appointment Spacing Model of care and associated factors among stable adult HIV clients on antiretroviral treatment Northwest Ethiopia 

Dear Dr. Telayneh:

I'm pleased to inform you that your manuscript has been deemed suitable for publication in PLOS ONE. Congratulations! Your manuscript is now with our production department. 

Kind regards, 

on behalf of

Dr. Hans-Uwe Dahms 

Academic Editor

PLOS ONE